# Polyglot or Not? Measuring Multilingual Encyclopedic Knowledge in Foundation Models

**Tim Schott, Daniel Furman, and Shreshta Bhat**[*]
School of Information
University of California, Berkeley
{timschott, daniel_furman, bhat_shreshta}@berkeley.edu

## Abstract

In this work, we assess the ability of foundation models to recall encyclopedic knowledge across a wide range of linguistic contexts. To support this, we: 1) produce a 20-language dataset that contains 303k factual associations paired with counterfactuals, 2) evaluate 5 models in a multilingual test, and 3) benchmark a diverse set of 24 models in an English-only test. Meta's LLaMA achieves the highest scores in both multilingual and English-only evaluations. Yet, an analysis of LLaMA's errors reveals significant limitations in its ability to recall facts in languages other than English, plus difficulties related to the location and gender of fact subjects. Overall, our findings suggest that today's foundation models are far from polyglots.[1]

## 1 Introduction

Can foundation models be used as multilingual knowledge bases? Foundation models typify an emerging paradigm that warrants further study; all-purpose Large Language Models (LLMs) that are trained on internet-scale corpora excel in generalization to some new tasks (Radford et al., 2018; Brown et al., 2020; Touvron et al., 2023). Their widespread adoption and ostensible credibility come with risks, though. For instance, foundation models inherit inaccuracies from training corpora (Argyle et al., 2023), which are in turn propagated downstream to the models that are fine-tuned from them (Bommasani et al., 2022; Chung et al., 2022). Additionally, foundation models spend the majority of their training phase absorbing information in English; for example, Touvron et al. (2023)'s LLaMA devotes two-thirds of its training dataset to an English-only subset of the Common-Crawl (Wenzek et al., 2020). Thus, foundation models are potentially deficient when performing non-English tasks (Kassner et al., 2021).

---

[*] All authors contributed equally
[1] Supporting code and data are openly released

## 2 Related Work

An impressive amount of knowledge is encoded within LLMs (Roberts et al., 2020), which store factual associations as key-value pairs within their memory (Geva et al., 2022; Meng et al., 2022b). Expose models to a large number of facts during self-supervised training, and they'll adeptly recall this information at deployment (Kaplan et al., 2020). However, along with useful facts, models can ingest dubious or harmful associations (Bender et al., 2021), particularly if training corpora are poorly constructed or unrepresentative of the world (Dodge et al., 2021).

To benchmark how robustly LLMs learn factual associations, Jiang et al. (2020) and Kassner et al. (2021) evaluated the encyclopedic knowledge of models like BERT (Devlin et al., 2019) and RoBERTa (Liu et al., 2019; Conneau et al., 2020) using rank-based approaches. Our study builds off this research in a few ways. We utilize a contrastive scoring approach, which tests the extent to which a model grasps a concept with more rigor than rank-based methods, as detailed below. Additionally, we inspect a diverse group of causal and masked language models rather than testing a single architecture to capture a more representative view of the field.

## 3 Task

We formulate the **Polyglot of Not?** test with cloze statements: given some context, we prompt an LLM to predict the next token. Factual associations are formalized as the triplet $\langle s, r, o \rangle$ where $s$ and $o$ denote the subject and object entity and $r$ is a linking relation, in line with Elsahar et al. (2018). Thus, the fact "Paris is the capital of France" is represented by $\langle Paris, capital\ of, France \rangle$ where "Paris" corresponds to $s$, "capital of" corresponds to $r$, and "France" corresponds to $o$. We then

prompt a model $M$ using the original natural language sentence with $o$ masked out. To assess if $M$ has correctly encoded an association, we calculate $P_M(o \mid s, r)$. Prior knowledge assessments employed a rank-based reward (Petroni et al., 2019) where a model is thought to understand the association if $o$ has a high chance of occurring as the next token (relative to all other options). However, this practice has pitfalls such as an inability to parse unsatisfactory outcomes for questions with numerous correct answers and a lack of insight into the LLM's confidence in its response. To address these issues, our work uses a variant of the Contrastive Knowledge Assessment (CKA) from prior work (Dong et al., 2022). Erroneous "counterfactuals" $\langle s, r, o' \rangle$ like $\langle Paris, capital\, of, Italy' \rangle$ are used to assess a model $M$'s understanding of $\langle s, r, o \rangle$. Simply put, if $M$ truly knows the fact, $P_M(o \mid s, r)$ should be larger than $P_M(o' \mid s, r)$ (Dong et al., 2022). Formally, CKA measures whether $M$ correctly knows a fact $\langle s, r, o \rangle$ via calculating:

$$\text{CKA}_\text{M}(s, r, o) = \frac{P_M(o \mid s, r)}{\mathbb{E}_{o'}[\, P_M(o' \mid s, r)\,]}$$

When $\text{CKA}_\text{M}(s, r, o) > 1$, the model is said to understand the factual association. This approach alleviates the issues that arise from ranking a model's vocabulary-wide token probabilities at inference; using counterfactuals elicits connections across different languages and contexts which forces the model to demonstrate generalized understanding of a given concept. Furthermore, examining the contrast allows us to quantify the confidence level with more nuance. Our work builds off Dong et al. (2022) by applying CKA to a multilingual dataset for the first time.

To carry out the test by language, we solicited cloze completions for each of the associations contained in the dataset. The percentage of fact-completions that $M$ recalls correctly is calculated by tallying up the number of completions where $\text{CKA}_\text{M}(s, r, o) > 1$ and dividing by the total number of completions. We accommodated different tokenizers by removing special tokens from text generation and ensuring that the completion probing corresponded to the first token to the right of cloze. Additionally, all evaluated models are fully open-source.[2] While we would have

liked to test proprietary LLMs such as GPT-4 (OpenAI, 2023), these models don't currently provide vocabulary token probabilities at inference, a prerequisite for CKA (see Assessing Open vs. Proprietary LLMs for details).

## 4 Dataset

The dataset includes 303k knowledge statements in 20 languages.[3] Each row includes: $dataset\_id$ (primary key), $stem$, $true$, $false$, $relation$, $subject$, and $object$. In total, the dataset contains 31 unique relation categories, 76,036 unique subjects, 18,837 unique objects, 18,503 unique trues, and 88,224 unique falses. Masked true/false objects consistently appear on the right-hand side of the statement to support masked and causal LLMs. The translated subset for each language contains different amounts of statements due to varying syntactic capacities to support this requirement. On average, a given fact appears in 12 of the 20 languages tested.

To construct the dataset, we first merged two English-language datasets from Dong et al. (2022) and Meng et al. (2022a) that share common lineage in the T-REx Wikidata (Elsahar et al., 2018) project. We then improved the dataset by filtering out inaccuracies and grammatical errors, as well as de-duplicating the $\langle s, r, o \rangle$ triplets, as detailed by Dataset Preprocessing in the Appendix. After preprocessing, the dataset contained 26,254 knowledge statements in English. We then used the Google Translate API to translate the data into 19 target languages: $bg$, $ca$, $cs$, $da$, $de$, $en$, $es$, $fr$, $hr$, $hu$, $it$, $nl$, $pl$, $pt$, $ro$, $ru$, $sl$, $sr$, $sv$, and $uk$ (ISO 639-1 codes). Our translation approach mirrors prior multilingual studies such as the programmatic translation of MMLU (Hendrycks et al., 2021) prompts when analyzing GPT-4. Additionally, work from Kassner et al. (2021) shows minimal practical differences when using machine versus manually translated cloze statements.

## 5 Results

Table 1 displays mean performance across the 20 languages used in the multilingual test. We present results for 5 foundational models here, with LLaMA-33B outperforming the others by a wide margin. We display LLaMA-33B's accuracy on

---

[2]LLaMA weights were accessed with Meta's permission

[3]https://huggingface.co/datasets/Polyglot-or-Not

| model | accuracy (%) |
|---|---|
| llama-33b | **79.31** (+/- 0.74) |
| m-bert | **62.00** (+/- 0.87) |
| bloom-7b1 | **57.70** (+/- 0.88) |
| xlm-roberta | **56.03** (+/- 0.90) |
| mt5-xl | **52.51** (+/- 0.91) |
| Random Baseline | 50 |

Table 1: **Multilingual test leaderboard**. Here, **accuracy** refers to the average performance of each model across 20 languages. The uncertainty estimates are averaged 95% confidence intervals computed from 10k bootstrap iterations per language. The results suggest tested models struggle to recall facts in a multilingual setting relative to English-only performance (Table 4).

| model | accuracy (%) |
|---|---|
| llama-65b | **89.56** (+/- 0.37) |
| llama-33b | **89.40** (+/- 0.38) |
| falcon-40b | **87.01** (+/- 0.41) |
| llama-13b | **86.66** (+/- 0.42) |
| llama-7b | **85.53** (+/- 0.43) |
| redpajama-7b | **85.07** (+/- 0.44) |
| Random Baseline | 50 |

Table 2: **English-only test leaderboard, top 6 models**. Here, **accuracy** refers to model performance on English data. The uncertainty estimates are 95% confidence intervals computed from 10k bootstrap iterations. Consistent with the trends in Table 1, LLaMAs of varying sizes emerge as the front-runners. Reference Table 4 in the Appendix for the full leaderboard.

each of the 20 languages individually in Table 3 and Figure 1. This model scores higher on languages written in Latin script than those written in Cyrillic script ($bg$, $ru$, $sr$, $uk$). A chi-squared test confirms that LLaMA-33B's performance is dependent on language script ($\chi^2 = 3570.58$, $p < 0.001$). Additionally, the results on the English-only test are displayed for two dozen models in Table 2 and 4. LLaMA models again top the leaderboard here, closely followed by Technology Innovation Institute's Falcon-40B (Penedo et al., 2023).

## 6  Analysis

### Training Data and Model Parameters

LLaMA excels in our tests relative to other foundation models. This challenges some previous notions that compute should be spent to support enormous (parameter-wise) models in lieu of larger

amounts of training data (Kaplan et al., 2020). For instance, LLaMA-7B with 1T tokens outperforms OPT-30B with 180B tokens (Zhang et al., 2022) on the English-only test (see Table 4). Moreover, the lean 110M parameter mBERT model (Devlin et al., 2019) outperforms two 7B parameter models on the multilingual test. Lastly, the LLaMA family provides a side-by-side comparison on the English-only test; the performance differential is largest from the 13B to 33B variants, aligning with the 1T to 1.4T training token jump (see Table 4).

### Subject Entity Error Analysis

We analyzed LLaMA-33B's errors across each of the 20 languages tested and found systemic gaps in its factual recall.[4] We began by exploring associations from our dataset that feature geographic locations as their subject entity. The 3,213 geographic entities we worked with appear in 48,606 prompts in our 20 language assessment (see Geographic Labeling in the Appendix for details). LLaMA-33B answered these types of questions correctly at an 89.94% clip. The top performing continent was Asia with 93.31% accuracy for 10,729 questions, and the lowest was Antarctica with 80.65% accuracy for 5,167 questions. A chi-squared test for independence comparing LLaMA-33B's performance on geographic questions related to Asian locations versus European locations confirms the superior performance on Asian locations is significant ($\chi^2 = 66.408$, $p < 0.001$).

We also explored whether LLaMA-33B's errors were systematically related to the gender (male/female) of a fact's subject. The 951 entities sampled appear in 16,003 prompts in the test (see Gender Labeling in the Appendix for details). LLaMA answered 75.87% of these questions correctly. Male subjects are nearly 5 times as common as female subjects in the sample, yet the model performs slightly worse on facts about male subjects. A chi-squared test for independence comparing LLaMA's performance on questions about male subjects compared to female subjects confirms its superior performance on facts about females is significant ($\chi^2 = 69.096$, $p < 0.001$).

---

[4]We analyzed LLaMA-33B because it both performs well on the multilingual test and boasts a parameter count suitable for interrogations on lightweight compute resources

## Wikipedia's Role in LLaMA Performance

LLaMA learns information by reading Wikipedia pages, so we studied data quality on each language's Wikipedia. We began by tabulating how many pages were present during LLaMA's training period (see Table 5). Of course, sheer page count is perhaps not the strongest indicator of the quality and diversity of information available on that language's Wikipedia; a single well-written page can be more informative than a dozen low-quality pages. To delve deeper, we analyzed Wikipedia pages from languages of interest (see Wikipedia Entity Analysis in the Appendix for details). Table 6 records word count, the number of named-entities that appear in the article (both total and unique), and the number of named subject entities in the dataset that appear in the article which we refer to as "target" entities (both total and unique). We adopt an approach such that a page that mentions 8 different target entities is considered to be denser and thus more informative than an article that narrowly focuses on a single target entity. Analysis of the articles we sampled reveals major gaps across each language's Wikipedia. We observe a strong and significant correlation (Pearson's $r = 0.78, p < 0.001$) between the average unique target entities on the page and LLaMA's performance; the more subjects on a Wikipedia page, the better LLaMA recalled facts in that language. This underlines the connection between dataset quality and performance on our assessment.

## Qualitative Insights

Qualitative analyses underscore the influence of frequency bias. For instance, LLaMA frequently erred when prompted with statements containing "Antarctica" in a variety of languages. In the English language prompt "Cape Monaco is a part of the continent of", LLaMA ranked "Europe" to be a more likely completion than the correct "Antarctica." Cape Monaco's Wikipedia page makes numerous references to European people and places (including its appellation), and LLaMA appears to prioritize the presence of a European entity rather than connect this location's correct continent. Not all signals in its training dataset, then, appear to be treated with equal diligence. What's more, when conducting pairwise comparisons between English and other languages for common facts, relative rankings remain largely consistent with overall performance. We observe degraded performance outside of English in LLaMA's results for prompts entailing English speaking countries, with Slavic languages exhibiting more significant deviations than others. Cross-lingual transfer of knowledge thus exhibits a lack of reliability.

## 7 Future Work

There are many directions left to pursue in this domain. Model weight editing in a multilingual setting presents a novel next step for our project since our data finds its roots in two projects (Dong et al., 2022; Meng et al., 2022a) that explore how to remedy inaccuracies located in LLMs. Also, applying the test to future open-source models will fortify this work's impact and relevance for future researchers (see Testing New Models in the Appendix for details). We can also add languages that use neither Cyrillic nor Latin scripts; we are working with native Hindi and Japanese speakers to create cloze statements in these languages. There is also work to be done regarding the variable difficulty of a given fact based on the availability of training data in that language; the values from our Wikipedia analysis could be used as prior probabilities in a future iteration of CKA. Additionally, we could analyze more facets of training corpora metadata. Perhaps it's possible to *causally* connect a model erring on a particular fact to artifacts in its training data rather than the measured, associative approach we adopt. Current work (Elazar et al., 2023) affords helpful scaffolding for this endeavor.

## 8 Conclusion

Here, we present a multilingual contrastive knowledge assessment of encyclopedic facts. Our original evaluation benchmarks 5 foundation models in a multilingual test and two dozen in an English-only test. Meta's LLaMA demonstrated superior performance in both settings. Accompanying analyses reveal that LLaMA struggles to operate in non-English languages, particularly in Cyrillic script, suggesting an absence of robust cross-lingual knowledge transfer. These findings vouch for the utility of high-quality, multilingual datasets for training the next-generation of foundation models. Our hope is that this project motivates future interrogations of foundation model data sources and provides a roadmap for others to conduct transparent evaluations. By doing so, LLMs can be better equipped for broad application across diverse linguistic contexts.

## Limitations

### Assessing Open vs. Proprietary LLMs

One prerequisite for carrying out the test is access to the full schedule of vocabulary-level token score probabilities generated when an LLM synthesizes text. For this reason, researchers in related inquiries typically work with fully open-source models with weights uploaded to the Hugging Face model hub (Jiang et al., 2020; Dong et al., 2022; Meng et al., 2022a). Proprietary models, meanwhile, lack this transparency rendering their generated texts resistant to analysis. Notably, OpenAI's GPT-3 API only surfaces the probabilities of the 5 most likely next tokens, a functionality which Hendrycks et al. (2021) leveraged to apply GPT-3 to their evaluation task. We submitted a request for this limit to be raised through OpenAI's official channel — a fully automated, chat-bot customer service agent — and we have yet to receive a response. What's more, the GPT-4 API nixed the reporting of token probabilities *entirely* (as of this writing), thwarting an important avenue for research into their newest foundation model and adding an additional layer of opacity into how their systems produce results (OpenAI, 2023). Likewise, as things stands today, the largest (parameter-wise) foundation models from other research consortiums such as DeepMind's Gopher (Rae et al., 2022), Google's LaMDA (Thoppilan et al., 2022), and Huawei's PanGu-Sigma (Ren et al., 2023) are all proprietary.

### GPU Resources

We performed experiments on a range of LLM families and sizes. This required many hundreds of hours of GPU usage (see Reproducibility in the Appendix for details). In total, we batched over 100 model runs that required approximately 500 hours of GPU usage. For instance, testing LLaMA-7B's performance on the 22,974 Portuguese factual associations in the dataset required 2.5 hours of GPU usage with 1x T4. In addition to having to schedule long-periods of compute uptime, we were also constrained by fixed resource requirements, using workstations with a single NVIDIA GPU. Thus, we could not evaluate the gamut of truly massive (parameter-wise) models in our experiments. Going forward, we believe more accommodations need to be made for groups to effectively experiment with LLMs, in particular as organizations release models that require extremely demanding compute requirements to host and run.

## Ethics Statement

Although we test a language model's ability to serve as multilingual knowledge bases, we do not find these models to be particularly reliable sources of knowledge; none of the models scored above 90% for any of the languages that we tested. We thus caution readers that LLMs should not be used as an authoritative source of facts — whether in a research setting such as this or in a real-world environment. The test sheds light on the types of languages, topics, and contexts where LLMs are more likely to produce factual errors, but the same methods might also enable a malicious actor to check whether a particular set of facts is committed to model memory and subsequently insert damaging information into a model that was not originally present in the training data with other methods, such as the MEMIT algorithm proposed by Meng et al. (2022b). Lastly, while our work points to the need for testing low-resource languages, the test at present is restricted to a relatively small number of languages (20), most of which are high-resource. We intentionally use the 20 languages included in the LLaMA training dataset in this work. However, future work must further explore fact-completion testing for low-resource languages and devote attention to a larger number of languages.

## Acknowledgements

We thank Professor David Bamman for helpful feedback and constructive suggestions. This project received funding from the School of Information at the University of California, Berkeley.

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

# Appendix

## Reproducibility

Supporting code and data are openly released on GitHub and Hugging Face, respectively. Text generation was conducted with the `transformers`[5] and `bitsandbytes`[6] packages (see Text Generation Configuration below for details). Subsequent steps of the **Polyglot or Not?** test were executed with the `pytorch`[7] package. In regards to compute resources, the experiments were performed on workstations equipped with various Nvidia GPUs. We employed 1x H100 (80 GB PCIe) for larger models (e.g., LLaMA-65B), 1x A100 (40 GB SXM4) for medium-sized models (e.g., LLaMA-33B), and 1x T4 (15 GB) for smaller models (e.g., LLaMA-13b/7b).

## Dataset Preprocessing

An in depth data preprocessing pipeline was applied to the dataset to improve its quality. The Calinet (Dong et al., 2022) dataset originally contained 50,451 stem/fact items which we consider "valid" cloze statements, items where the masked object appears on the right-hand side of the stem. Many of these stem/fact pairings were paraphrased, though, to support their model rewrite process, which this paper does not explore. After removing these paraphrased $\langle s, r, o \rangle$ triplet duplicates, we were left with 11,960 statements from this data pool. Meanwhile, the ROME (Meng et al., 2022a) dataset contributed 21,919 valid stem/fact pairs, all of which were unique $\langle s, r, o \rangle$ triplets. We merged the data and were left with 33,870 items. From there, we performed the following enhancements:

- Removed 227 stem/fact pairs that were manually flagged as errors

- Removed 371 stem/fact pairs with "a/an + _" due to consistent grammatical errors

- Removed 3,088 stem/fact pairs where the correct fact is explicitly stated in the stem, rendering the completion trivial

- Removed 610 stem/fact pairs that were relation P190 (sister city) due to consistent inaccuracies

[5] https://github.com/huggingface/transformers
[6] https://github.com/TimDettmers/bitsandbytes
[7] https://github.com/pytorch/pytorch

- Removed 418 stem/fact pairs that were relation P140 (religion) to filter sensitive topics

- Removed 490 stem/fact pairs that were relation P530 (diplomatic ties) due to consistent inaccuracies

- Removed 1,427 stem/fact pairs that were relation P27 (citizen of) due to consistent inaccuracies

- Removed 576 stem/fact pairs that were relation P463 (affiliated with) due to consistent inaccuracies

- Removed 39 stem/fact pairs that compared football with soccer due to cultural differences in these word meanings

- Removed 131 stem/fact pairs with "expired at" wording due to awkward phrasing

- Removed 50 stem/fact pairs with "-language" wording due to awkward phrasing

- Removed 73 stem/fact pairs with facts/counterfacts starting with "the" due to the frequency of the word "the" in training datasets

- Removed 125 stem/fact pair duplicates to retain a dataset of entirely unique $\langle s, r, o \rangle$ triplets

Our straightforward improvements provide more validity to our pool of data and its ultimate use in the **Polyglot or Not?** test, such as removing the over 3,000 statements whose correct answer can be found in the unmasked portion (bullet number 3). See Table 7 for a handful of examples filtered out during the above operations. After preprocessing, we are left with 26,254 unique rows in the final English-only subset of our dataset.

## Geographic Labeling

We sought a labeled dataset of geographic entities connected to the continents they're located on. To do so, we filtered our original dataset down to the Wikidata relation IDs that most clearly signal that a geographic entity, such as Paris or France, occupies the subject of the stem: capital (relation P17 + P1376), continent (P30), country (P36), shares border with (P47), and is in the territory of (P131). Then, we extracted the unique, English translations of the subjects from this data, leaving us with 3,427 "geographic"

entities in our dataset. To more quickly move into substantive analysis, we utilized a Generative AI assistant, ChatGPT (gpt-3.5-turbo accessed April, 2023)[8], to label these entities by geographic continent. Our prompt (see below) offered an option for an "unsure" label if the assistant did not know the correct answer, the location stretched across multiple continents, etc. Of the 3,427 we requested prompts for, the assistant labeled 3,213 with a tag for one of the world's continents. To verify the veracity of the labels we randomly sampled 10% of the labeled data and found that the affixed continent labels were correctly applied to every entity in the validation sample. The resulting labeled data provided interesting terrain for mining insights, as detailed in the Subject Entity Error Analysis subsection. Prompt used:

```
I have a list of locations. Can
you return the continent on which they
are located in the following format:

Iran|AS
Bavaria|EU
Pennsylvania|NA

If there are items in the list that
don't seem like locations or perhaps are
very difficult to classify you can write
"unsure" beside those, e.g.

WTJU-FM|unsure
Ottoman Empire|unsure
```

### Gender Labeling

We also desired a labeled dataset of person entities connected to their assigned birth gender, as understood in the popular consciousness. To do so, we filtered our original dataset down to the Wikidata relation IDs that most clearly signal that a person entity, such as Steve Jobs or Marie Curie, occupies the subject of the stem: place of death (relation P20), position held (P39), field of work (P101), native language (P103), occupation (P106, employer (P108), position played on team (P413), sport (P641), work location (P937), and instrument (P1303). We followed a near-identical procedure for Gender Labeling as we did for Geographic Labeling, using ChatGPT to label

these identities by gender. However, because there are far more people entities in our dataset after filtering for these relation IDs (7,905 in total) we randomly sampled a portion of them, extracting 1,200 unique entities to hand off to ChatGPT. Our prompt (see below) for gender also offered an option for an "unsure" label if the assistant did not know the correct answer, the entity wasn't a name, etc. Of the 1,200 we requested prompts for, the assistant labeled 1,057 with a gender tag. To verify the veracity of the labels we randomly sampled 10% of the labeled data and found that the affixed gender labels were correctly applied to every entity in the validation sample. The resulting labeled data is also explored in the Subject Entity Error Analysis subsection. Prompt used:

```
I have a list of names. Can you
return the gender (male, female, or
other) in the following format:

Sundar Pichai|Male
Brigitte Fontaine|Female

If there are items in the list that
don't seem like names or perhaps are very
difficult to classify, you can write
"unsure" beside those, e.g.

WTJU-FM|unsure
Wagnerian|unsure
```

### Wikipedia Entity Analysis

To produce Table 6, we began by randomly sampling 10k pages from every language of interest's Wikipedia via Wikipedia's REST API. We did this because we wanted to gather a sample of the text in the article body for each language. From there, we extracted the body content of these articles and performed minimal preprocessing such as removing citations and navigation headers. With the clean page content in hand, we then used a named-entity recognition utility from SpaCy.[9] SpaCy provides models for 14 of the 20 languages LLaMA was tested on. For each of these languages, the `core_news_lg` tagger was used save for English where we used the `core_web_lg` tagger. We then tallied counts for the entities found in each page. We tracked the overall and distinct number of entities found in

---

[8]https://chat.openai.com

[9]https://spacy.io

each article. We also stored the overall and distinct subset of entities that are found in each article and who appear in our dataset.

**Text Generation Configuration**

All tests were conducted with the same text generation hyper-parameters, by and large employing the default configuration from the `transformers` package. The principle deviation from the default settings in our tests was the use of mixed-precision quantization; we explored the impact of adjusting matrix multiplication precision on a given model's test performance to confirm the efficacy of this method. Specifically, we ran LLaMA-7B and LLaMA-13B on the English-only subset of the dataset under both $fp16$ and *8-bit* configurations. In the case of $fp16$ precision, all values were simply assigned the $torch.float16$ data type. For *8-bit* precision, we adopted the mixed-precision algorithm from the $bitsandbytes$ package as presented by Dettmers et al. (2022), which utilizes the $torch.int8$ data type for the majority of the values and the $torch.float16$ data type for outliers. We found minute but noticeable differences in model performance between the two precision levels (0.35-0.47%, see Table 8). The savings in GPU memory consumption, however, were much more significant by comparison. By opting for *8-bit* over $fp16$ precision, we reduce the memory footprint of the two models roughly in half. Based on these results, we determined that the trade-offs between performance and memory footprint were acceptable for our test, as we were running tests on relatively lightweight compute resources. We thus elected to employ *8-bit* precision throughout the experiments.

**Testing New Models**

The results included herein exclusively feature foundation models released before June 2023. We have continued to test new LLM releases since then, including Meta's Llama-2 model family, Mistral.ai's Mistral-7B, and TII's Falcon-180B. A regularly updated leaderboard is maintained at the project repo, with the hopes that the **Polyglot or Not?** test retains its relevance and impact as text-based foundation models proliferate.[10]

---

[10]https://github.com/daniel-furman/polyglot-or-not

| language | accuracy (%) | pairs |
|---|---|---|
| English | **89.40** (+/- 0.38) | 26,254 |
| German | **85.74** (+/- 0.53) | 16,287 |
| Dutch | **85.35** (+/- 0.46) | 25,590 |
| Italian | **84.39** (+/- 0.49) | 20,448 |
| French | **84.18** (+/- 0.52) | 18,395 |
| Swedish | **84.06** (+/- 0.48) | 21,576 |
| Catalan | **84.01** (+/- 0.52) | 18,898 |
| Portuguese | **83.81** (+/- 0.48) | 22,974 |
| Romanian | **82.72** (+/- 0.57) | 17,568 |
| Danish | **81.79** (+/- 0.50) | 23,365 |
| Spanish | **81.74** (+/- 0.57) | 18,786 |
| Czech | **77.94** (+/- 0.84) | 9,427 |
| Polish | **77.50** (+/- 0.84) | 9,484 |
| Croatian | **76.69** (+/- 0.97) | 7,358 |
| Slovenian | **75.99** (+/- 0.95) | 7,873 |
| Hungarian | **75.74** (+/- 1.24) | 4,650 |
| Ukrainian | **73.00** (+/- 0.98) | 7,918 |
| Bulgarian | **72.50** (+/- 0.61) | 20,577 |
| Russian | **69.72** (+/- 1.57) | 3,289 |
| Serbian | **60.01** (+/- 1.30) | 5,426 |
| Random Baseline | 50 | - |

Table 3: **LLaMA-33B's performance across languages**. Here, **accuracy** denotes the LLaMA-33B model's performance assessed individually for each language, while **pairs** refers to the number of stem/fact items evaluated per language. LLaMA-33B demonstrates higher proficiency with languages utilizing the Latin script as compared to those using the Cyrillic script (Ukrainian, Bulgarian, Russian, and Serbian). A chi-squared test substantiates a significant dependency of the model's test performance on the language script ($\chi^2 = 3570.58, p < 0.001$). For a graphical representation of these results, refer to Figure 1 below.

| model | accuracy (%) | params | *n* tokens |
|---|---|---|---|
| llama-65b | **89.56** (+/- 0.37) | 65.2B | 1.4T |
| llama-33b | **89.40** (+/- 0.38) | 32.5B | 1.4T |
| falcon-40b | **87.01** (+/- 0.41) | 40B | 1T |
| llama-13b | **86.66** (+/- 0.42) | 12.5B | 1T |
| llama-7b | **85.53** (+/- 0.43) | 6.7B | 1T |
| redpajama-7b | **85.07** (+/- 0.44) | 7B | 800B |
| mpt-7b | **83.39** (+/- 0.46) | 7B | 1T |
| opt-30b | **82.09** (+/- 0.47) | 30B | 180B |
| redpajama-3b | **82.09** (+/- 0.47) | 3B | 800B |
| opt-13b | **81.94** (+/- 0.46) | 13B | 180B |
| gpt-neox-20b | **81.50** (+/- 0.47) | 20B | 420B |
| falcon-7b | **81.34** (+/- 0.47) | 7B | 1.5T |
| gpt-j-6b | **81.14** (+/- 0.47) | 6B | 420B |
| pythia-12b | **80.53** (+/- 0.48) | 12B | 420B |
| t5-v1-xxl | **76.55** (+/- 0.52) | 13B | 34B |
| bloom-7b1 | **76.16** (+/- 0.51) | 7.1B | 341B |
| gpt2-xl | **73.76** (+/- 0.54) | 1.5B | - |
| bert | **72.60** (+/- 0.54) | 110M | - |
| m-bert | **71.80** (+/- 0.55) | 110M | - |
| stablelm-7b | **68.85** (+/- 0.55) | 7B | 1.5T |
| distilgpt2 | **64.23** (+/- 0.59) | 82M | - |
| mt5-xxl | **61.58** (+/- 0.59) | 13B | - |
| xlm-roberta | **61.55** (+/- 0.59) | 355M | 295B |
| mt5-xl | **59.96** (+/- 0.59) | 3.7B | - |
| Random Baseline | 50 | - | - |

Table 4: **English-only test leaderboard**. Here, **accuracy** refers to model performance on English data. The uncertainty estimates are 95% confidence intervals computed from 10k bootstrap iterations. **Params** and *n* **tokens** record each model's number of parameters and number of dataset tokens, respectively (when such data is available). Consistent with the trends in Table 1, LLaMAs of varying sizes emerge as the front-runners.

| language | article count |
|---|---|
| English | 6,513,291 |
| German | 2,698,267 |
| Swedish | 2,551,218 |
| French | 2,430,636 |
| Dutch | 2,092,862 |
| Russian | 1,828,011 |
| Spanish | 1,782,912 |
| Italian | 1,758,843 |
| Polish | 1,525,414 |
| Ukrainian | 1,160,183 |
| Portuguese | 1,093,217 |
| Catalan | 702,281 |
| Serbian | 659,580 |
| Hungarian | 505,754 |
| Czech | 505,105 |
| Romanian | 431,067 |
| Bulgarian | 282,130 |
| Danish | 280,923 |
| Croatian | 212,088 |
| Slovenian | 176,565 |

Table 5: **Wikipedia page counts**. The number of articles available on Wikipedia during LLaMA's training time period of June 2022, as reflected by the article count for each language surfaced on archive.org (arranged descending by article count). Even a high-resource language like Romanian possesses a rather small Wikipedia in comparison to other languages like French. (The corresponding archive.org URLs, which link to the initial archived copy of the language's homepage on or as close as possible to June 15th, 2022 can be found in our codebase.)

| language | words | entities | unique entities | targets | unique targets | accuracy (%) |
|---|---|---|---|---|---|---|
| Catalan | 384.43 | 26.51 | 19.54 | 3.82 | 2.31 | 84.01 |
| Croatian | 273.29 | 24.98 | 19.68 | 0.87 | 0.60 | 76.69 |
| Danish | 255.77 | 22.71 | 16.90 | 3.30 | 2.07 | 81.79 |
| Dutch | 156.54 | 25.25 | 19.78 | 1.85 | 1.28 | 85.35 |
| English | 463.15 | 70.12 | 50.19 | 6.86 | 3.68 | 89.40 |
| French | 491.93 | 36.82 | 26.27 | 4.49 | 2.58 | 84.18 |
| German | 418.59 | 40.95 | 30.46 | 3.90 | 2.40 | 85.74 |
| Italian | 391.67 | 31.09 | 21.52 | 3.96 | 2.24 | 84.39 |
| Polish | 214.29 | 32.70 | 25.74 | 0.90 | 0.60 | 77.50 |
| Portuguese | 320.44 | 24.95 | 17.86 | 3.60 | 2.18 | 83.81 |
| Romanian | 231.10 | 42.72 | 33.68 | 2.39 | 1.62 | 82.72 |
| Russian | 382.84 | 35.70 | 26.35 | 0.86 | 0.58 | 69.72 |
| Spanish | 470.56 | 33.04 | 23.80 | 4.25 | 2.41 | 81.74 |
| Swedish | 95.03 | 9.37 | 7.42 | 1.30 | 0.95 | 84.06 |
| Ukrainian | 283.64 | 25.32 | 19.90 | 1.21 | 0.80 | 73.00 |

Table 6: **Wikipedia content analysis**. Results of performing named-entity recognition on a random sample of 10k Wikipedia articles across 15 languages (arranged alphabetically by language name). Reported metrics correspond to per-page averages: **words** is the article word count as reported by SpaCy's language specific tokenizer. **Entities** and **unique entities** represent the total and distinct entity counts, respectively, from SpaCy's named-entity recognition tagger on the page text while the **targets** and **unique targets** columns correspond to the counts of entities that occupy the subject position of stems in our dataset. LLaMA's test **accuracy** for each language occupies the right-most column, as is also displayed in Table 1 and Figure 1. We find that LLaMA's performance is significantly correlated with the number of unique target entities found in our sampled pages (Pearson's $r = 0.78, p < 0.001$). Other takeaways include the rather low average word count of articles on Swedish language Wikipedia due to its high proportion of machine generated pages.

| stem | true | false(s) | notes |
|---|---|---|---|
| "Porsche Panamera is developed by" | "Porsche" | "BMW" | Answer in **stem** |
| "Vincent van Gogh took up work in" | "The Hague" | ["Belfast", "Worpswede"] | "The" in **true** |
| "Muhammad is follower of" | "Islam" | "Buddhism" | Religion relation |

Table 7: **Examples of data filtered out by preprocessing**. Here, we show a small sample of items that were filtered out by the preprocessing pipeline, with steps detailed in Dataset Preprocessing above. The first and third items originate from Meng et al. (2022a) while the second item originates from Dong et al. (2022).

| model | precision | accuracy (%) | memory footprint (GB) |
|---|---|---|---|
| llama-13b | $fp16$ | **87.01** (+/- 0.40) | 26.2 |
| llama-13b | $8\text{-}bit$ | **86.66** (+/- 0.42) | 14.5 |
| llama-7b | $fp16$ | **86.00** (+/- 0.42) | 14.7 |
| llama-7b | $8\text{-}bit$ | **85.53** (+/- 0.43) | 8.3 |

Table 8: **Quantization experiments for LLaMA-7B and LLaMA-13B**. Here, **accuracy** denotes the model's performance on English-only data. A small dip in accuracy (0.35-0.47%) is observed between $fp16$ and $8\text{-}bit$ precisions.

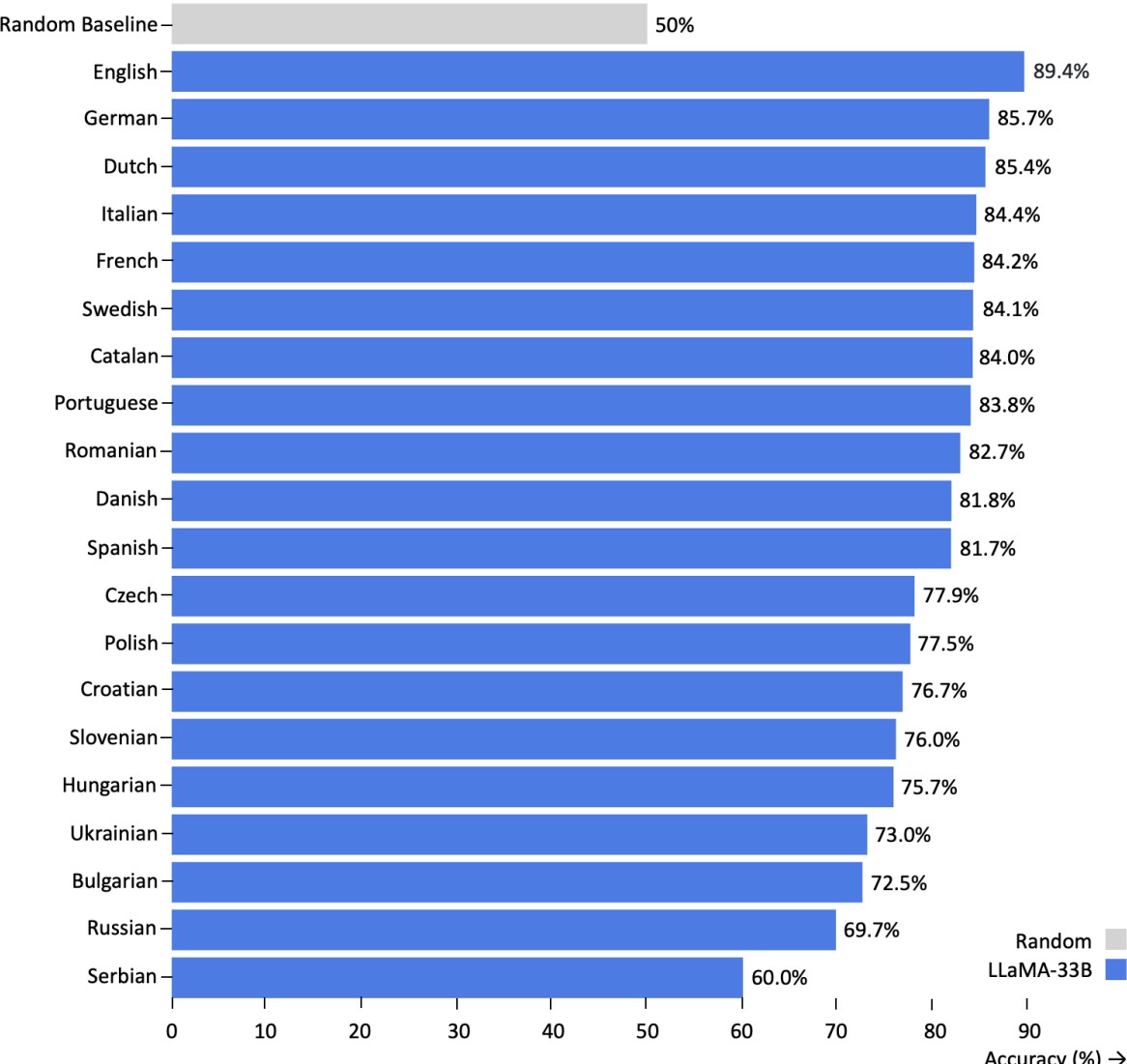

Figure 1: **LLaMA-33B's performance across languages, visualized.** The model (blue) scores higher on languages written in Latin script than those written in Cyrillic script (Ukrainian, Bulgarian, Russian and Serbian). A chi-squared test confirms that the model's test performance is dependent on language script ($\chi^2 = 3570.58, p < 0.001$). For a tabular representation of these results, refer to Table 3 above.