# OpenReview forum: "Polyglot or Not? Measuring Multilingual Encyclopedic Knowledge in Foundation Models"
_EMNLP/2023/Conference — EMNLP 2023 Main_

### Official Review · Reviewer_vbS2 · 2023-07-21

**Soundness:** 4

**Excitement:**

4: Strong: This paper deepens the understanding of some phenomenon or lowers the barriers to an existing research direction.

**Paper Topic And Main Contributions:**

This paper evaluates Encyclopedic Knowledge Retrieval from LLMs using a manually curated dataset spanning some 20 languages. The finding is clear, that there is a decline according to language (and/or script --- though this distinction is not really within scope of the paper) so even the most powerful LLMs with lots of parameters can't really be said to abstract (language-independent) knowledge, rather they abstract a combo of language independent knowledge and language-specific patterns.

**Questions For The Authors:**

Fact knowledge is formalized as ⟨s, r, o⟩ triples (said to be from an "original natural language sentence") whereas the dataset is rather like (stem, subject, object) triples where stem is a grammatically valid formulation of a fact-triple. There's some philosophical or modeling clarification needed here. Are fact triples language independent, e.g., is (Paris, capital of, France) the same fact as (Paris, capitale de, France) and (Paris, Hauptstadt von, Frankreich)? Several different (stem, subject, object) triples can represent the same fact as there are usually several different ways to formulate things with the same propositional content.

**Reasons To Accept:**

It's a well conducted study and the finding is clear and of interest.

**Reasons To Reject:**

There are no real reasons to reject but of course some aspects could have been done differently or ontop. The accuracy numbers will change if the prior/baseline ratio of 50% true/false questions are asked. For many real world applications the prior will not be 50/50. Would have been interesting to know how drastic the change in accuracy would be.

**Reproducibility:**

5: Could easily reproduce the results.

**Reviewer Confidence:**

3: Pretty sure, but there's a chance I missed something. Although I have a good feel for this area in general, I did not carefully check the paper's details, e.g., the math, experimental design, or novelty.

**Typos Grammar Style And Presentation Improvements:**

There are broken hyperlink references, e.g.

fn 13
https://github.com/MicrosoftDocs/azuredocs/
blob/main/articles/cognitiveservices/
openai/includes/chat-markup-language.md

and the link refs do not have dates of access.

---

> ### Author Rebuttal · Authors · 2023-08-28
>
> Thank you for your comments! We are appreciative of the time you spent learning about our project, where we explore the capacity for foundation models to retrieve encyclopedic knowledge and show that even massive LLMs are hindered in their ability to learn language-independent knowledge. As you neatly put it: foundation LLMs appear to “abstract a combo of language-independent knowledge and language-specific patterns”, as uncovered when they are tasked with a multilingual cloze completion task like our “Polyglot or Not” test.
>
> You’re right to point out that our test takes advantage of comparing true/false completions. The point about how in **real world application the prior will not be 50/50**, from our perspective, warrants follow-up. We employed a contrastive knowledge assessment test. At the heart of the method is the comparison between a ground truth “fact” and one or more “counterfactuals.” As is evident in our dataset, many of the items have multiple counterfactuals. Take an illustrative example: “Steve Jobs was the founder of ____.” The true completion would be “Apple” and the counterfactuals may be a list of several false completions like “Google”, “Amazon”, and “Facebook”. Our test compares the probability of the true prediction against the mean probability of the false predictions. In this manner, we take into account a more nuanced perspective on our cloze test than a black and white 50% prior/baseline ratio to probe a particular piece of knowledge. The test thus necessitates the comparison of a factual completion with counterfactual completion(s).
>
> To answer your question about **language independence**: our dataset intentionally avoids including multiple variations of a fact to prevent double-counting. We utilize relation IDs to ensure that only one standard form of a given triplet is included, such as "is capital of." In the pre-processing stage, we implement this de-deduplication process for the English split of the dataset. This set is retained during translation across the 19 non-English languages in the test, further solidifying the language-independent representation of facts within our dataset. To reiterate, the statements with identical <s,r,o> triplets across languages are indeed treated as equivalent, representing a single piece of knowledge. We will include these details elaborating our design choice and its implications.
>
> Lastly, thank you for pointing out the fixes we can make to the broken hyperlink reference in fn 13. It appears the structure of Microsoft’s “azure-docs” repo has been re-factored since our original date of submission. We will add the updated link to our paper along with dates of access to all links in the paper’s footnotes to ensure that we are accurately documenting these external sources in a way that’s visible for future researchers.
>
> We hope that our rebuttal has clarified some of the strengths of our project and demonstrated our intention to mend any potential shortcomings therein.

---

### Official Review · Reviewer_CaEc · 2023-08-03

**Soundness:** 3

**Excitement:**

3: Ambivalent: It has merits (e.g., it reports state-of-the-art results, the idea is nice), but there are key weaknesses (e.g., it describes incremental work), and it can significantly benefit from another round of revision. However, I won't object to accepting it if my co-reviewers champion it.

**Paper Topic And Main Contributions:**

This paper describes experiments that assess how well LLMs retrieve encyclopedic knowledge. The experiments are multilingual.
The authors evaluate 5 LLMs on 20 languages. They build off work done by Dong et al. 2022 for English. The main conclusion is that LLMs do not perform as well on non-English datasets, especially those written in non-Latin scripts (Cyrillic).


**Questions For The Authors:**

I understand that the original dataset was in English and then it was translated into multiple languages. However, some facts might culture/language-specific. Something that is common knowledge in one culture, might be not necessary so in another (of course, I"m not talking about such basic facts as "capital of"). Did you control for it?

It would be interesting to include cross-lingual (rather than multilingual) experiments. Is it possible to for the model to predict o in language X given s and r in language Y? What language pairs work better? (Latin-based or Cyrillic- based?)

**Reasons To Accept:**

This is an interesting multilingual exploration on how LLMs perform in non-English settings. The paper also produced multilingual datasets for this task useful for further experimentation.
The paper includes an interesting section on subject entity error analysis.


**Reasons To Reject:**

My main criticism comes from the fact that apart from reporting the results of the experiments, the paper doesn't attempt to explain "why" they got these results. This is the most interesting question here.

**Reproducibility:**

4: Could mostly reproduce the results, but there may be some variation because of sample variance or minor variations in their interpretation of the protocol or method.

**Reviewer Confidence:**

3: Pretty sure, but there's a chance I missed something. Although I have a good feel for this area in general, I did not carefully check the paper's details, e.g., the math, experimental design, or novelty.

---

> ### Author Rebuttal · Authors · 2023-08-28
>
> Thank you for your remarks! It’s very meaningful that our project received insightful feedback from another member of the field. The goal of our project was to interrogate the capacity of LLMs to retrieve factual information in both an English-only and multilingual setting. Through our original “Polyglot or Not” benchmark and supporting analysis, we ultimately conclude that today’s "foundation models" – a term commonly used in the literature (see Section 1, Introduction) – possess numerous shortcomings when tasked with retrieving encyclopedic knowledge in languages other than English.
>
> As you point out, most prior work in this domain is conducted in an English-only setting. As nearly 3 out of 4 people worldwide do not speak or write fluently in English, it’s critical to introduce verifiable LLM benchmarks that evaluate their capacity to work in languages other than English. In our view, this core focus of our project ties in nicely with the EMNLP ‘23 theme-track: *While the new generation of Large Language Models such as GPTX, LLAMA, BLOOM etc. claim to perform at unprecedented levels for generation and understanding, we are in unexplored territory on many aspects of such LLMs.* This paper represents one such “exploration” into that territory and provides a level-headed view of LLM performance via a novel benchmark. Along the way, we broaden our focus to other areas of interest relevant to the NLP community like equity and fairness through our supporting analysis.
>
> We appreciate your recognition that the multilingual dataset we’ve developed is highly suitable for use in future research. We hope that by disseminating the dataset in multiple formats – as a file in our Github repository, but also on the easily viewable open-source repository “Hugging Face” – more people will be able to work with and ultimately use our pool of data for future research.
>
> We agree that the **why** of these results is one of the many interesting questions borne out by our work. To start, we’ll point out that this is perhaps the fundamental outstanding question in language modeling writ large. As downstream users of these widely disseminated ‘foundation’ models, we’re often left wondering what specific modeling decisions or dataset quirks caused a model to generate one piece of text or another. Our paper strives to contribute to this discourse by conducting a comprehensive battery of analysis.
>
> That said, we concede that we do not make explicitly causal claims in our study. We believe this cautious approach avoids assertions that lack proper empirical grounding. We do provide citations to an NLP survey (see Section 7, Future Work) that focuses on the causal components of fact retrieval catering to readers interested in deep study on this subject matter.
>
> Instead, in this short paper, we approach this “why” question through numerous types of analysis. In their totality, we believe that the quantitative and qualitative insights included in our study go further than mere reportage. Let’s recall some of those experiments and how their results connect back to the question you pose of “why” we see the results we do.
>
> We conduct an analysis (see Section 6, Wikipedia’s Relationship to LLaMA’s Performance) of Wikipedia’s ‘density’ and establish a strong and significant correlation (Pearson’s r = 0.78,p < 0.001) between the presence of subjects from our dataset on Wikipedia pages and LLaMA’s ability to recall facts in that language. This demonstrates the impact of high-quality, dense training data for imbuing LLMs with credible factual associations. This work also provides original commentary about a ubiquitous source of training data for foundation models. To continue, we also detail the impact of training data corpus size (see Section 6, Training Data Versus Model Parameters). These findings provide a tangible "why" behind the differences in performance that we observed and strongly link our paper to other contemporary researchers in the space. The qualitative arm of the paper (see Section 6, Qualitative Insights) also elucidates our assessment results while avoiding spurious claims of causality. For instance, we note that entity bias likely hampers LLaMa’s ability to accurately answer questions about geographic locations that have European-influenced names or origins, such as Antarctica’s Cape Monaco, but are not actually located on the European continent. **For each of these and other experiments we can more clearly define their connection to the "why" of our benchmark results within Section 6, Analysis.**
>
> You are right to point out that some facts are culture/language-specific. Take a multilingual model that has been trained on Swedish language Wikipedia data. We could prompt this model to complete a cloze statement regarding an American NFL player. Intuitively, one would surmise there is less high-quality information about that specific player and organization on the Swedish Wikipedia compared to the English language wikipedia.  Thus, this question will be more “difficult” when posed in Swedish compared to English. We do not control for this sort of phenomenon at this time in our paper, because it would be quite challenging to formulate a fair and balanced process for doing so. Indeed, what is culture/language-specific for one stem/fact pair might be universal for another culture or language and changes constantly based on new types of interactions between different cultures. Challenges notwithstanding, given the work we have already carried out, it is possible that the summary statistics from our Wikipedia analysis (see Table 6, Wikipedia content analysis) could be incorporated into as ‘priors’ in our assessment to down/upweight the raw likelihoods returned from prompting the model. This is one way we could control for culture/language specific knowledge during our assessment. **We can discuss this idea in our Future Work section (with the caveats being that, as mentioned, formulating this prior in an even-handed fashion would be somewhat challenging).**
>
> If we are interpreting your remarks about cross-lingual experimentation correctly, a summary of the process would look like: part of the sentence is in one language and the output should be in another. This would be a challenging, but also highly interesting, modification to our existing methodology, because it implicates the universality of internal fact representations - and if/how linguistic similarities play a role in it. **This is certainly a direction we could delve further into in our Future Work section as well.**
>
> We hope that this rebuttal clarified the strengths of our project and shows you that we are committed to improving this paper even further thanks to your insightful remarks.

---

### Official Review · Reviewer_JE4L · 2023-08-05

**Soundness:** 4

**Excitement:**

4: Strong: This paper deepens the understanding of some phenomenon or lowers the barriers to an existing research direction.

**Paper Topic And Main Contributions:**

In this short paper, the authors measure the performance of causal and masked language models when it comes to retrieving encyclopedic knowledge encoded as factual triples <s,r,o>. The analysis is done for English-only as well as multilingual settings (EN data is automatically translated to 19 other languages) with different sets of models. Results show that LLaMA is consistently more accurate when compared to other models, with EN results being consistently better. Authors perform several analyses to compare performance against number of parameters, for geographical entities/subjects, for persons w.r.t. gender, against entity density within Wikipedia articles, which serves as training data for LLaMA.

**Questions For The Authors:**

* Quite a few stem/facts pairs are removed from the merged dataset. The reasons are stated in the appendix. How were these assessments (e.g., "consistent inaccuracies") made? Manually? On a sample? The article would benefit from including some pairs for some of these categories.

**Reasons To Accept:**

* Use of contrastive learning and previously-proposed contrastive knowledge assessment  score where previous approaches used rank-based methods
* Evaluation of 24 models (EN only) and 5 models (multilingual)
* Development of a dataset with 303,000 <s,r,o> facts in 20 languages
* Analysis of relationship between LLaMA performance and counts of words/entities identified in Wikipedia articles
* Code and data released to the public

**Reasons To Reject:**

* The effort detailed here makes use of existing datasets, previously-proposed methods/scores, and existing models.

**Reproducibility:**

5: Could easily reproduce the results.

**Reviewer Confidence:**

4: Quite sure. I tried to check the important points carefully. It's unlikely, though conceivable, that I missed something that should affect my ratings.

**Typos Grammar Style And Presentation Improvements:**

* Please use commas to make large numbers more readable (e.g., 76036 -> 76,036) throughout the article

---

> ### Author Rebuttal · Authors · 2023-08-28
>
> Thank you for your thoughtful comments! We appreciate the time you spent with our project, where we focus on evaluating how well foundation models retrieve encyclopedic knowledge in a multilingual setting. We believe our original “Polyglot or Not” assessment and numerous supporting analyses enhance the research community’s understanding of how LLMs retrieve factual knowledge and even possess insights generalizable to the broader public.
>
> We’d like to highlight that the particular class of multilingual models we tested are referred to in the literature as ‘foundation models’ (see Section 1, Introduction). This is an important group of models which are typically used as base models for fine-tuning and therefore carry large downstream impacts. The rationale behind selecting these 5 models is important, and relates to your note about **existing models**. These are models that are widely used in research and industry settings and thus warrant close inspection: What do they do well? What are their drawbacks? And so on. Testing 5 multilingual foundation models across 20 different languages creates the conditions for fascinating comparisons.
>
> Helpfully, this core focus of our project ties in nicely with the EMNLP ‘23 theme-track: *While the new generation of Large Language Models such as GPTX, LLAMA, BLOOM etc. claim to perform at unprecedented levels for generation and understanding, we are in unexplored territory on many aspects of such LLMs.* This paper represents one such “exploration” into that territory and provides a level-headed view of LLM performance via a novel benchmark. Along the way, we broaden our focus to other areas of interest relevant to the NLP community like equity and inclusion – as you note in your review, *compare performance against number of parameters, for geographical entities/subjects, for persons w.r.t. gender, against entity density within Wikipedia articles, which serves as training data for LLaMA.*
>
> To continue, we’d also like to respond to your mention of **existing methods/scores**. Our project leverages a scoring heuristic from prior research in truly new ways. We present the first multilingual, contrastive knowledge assessment for LLMs (see Section 3, Task) applied via a multilingual cloze completion task. Understanding the significance of both prongs of this assessment – multilingual and contrastive – is critical for interpreting our results and understanding their broader significance. Many knowledge retrieval projects only work with information in English. Given that over 75% of the world’s population doesn’t speak or write in English, and these models are ostensibly multilingual, we found it pertinent to expand our knowledge assessment beyond English language factual associations. The contrastive portion is also important to underline. Many assessments simply use a rank-based approach to score LLM factual retrieval. We detail the downsides to this in our paper (see Section 3, Task) such as a bias towards tokens that appear frequently in the dataset. Our decision to use an existing method was deliberate, aimed at focusing on a multilingual assessment where the existing method already provided a robust foundation. The synergistic use of these two elements allows for new insights that would not have been possible using either alone.  **Your note motivates us to clarify why our work is differentiated from prior work as part of Section 3.**
>
> Your point about using **existing datasets** warrants a small follow-up. As noted in your summary, we ultimately incorporate those into a new *dataset with 303,000 <s,r,o> facts in 20 languages.* We amalgamated two English-only data pools into a well-curated collection of data that spans 20 different languages. We removed inaccurate/ambiguous/biased facts, and maintained consistency across translations. In doing so we hope to support future researchers. One of the ways we make good on this is by making our dataset available in multiple formats, including on Hugging Face. You can view, search and then download our benchmark data right from your browser. We believe this empowers future researchers and members of the general public alike to easily carry out follow-up projects leveraging our pool of data. The interest in the dataset we’ve put together has been resoundingly positive. In the month of May, for instance, it received over 500 downloads!
>
> With respect to the “consistent inaccuracies” that we mention in our Appendix, a bit of extra explanation is needed here (and can be added to our paper in the Dataset Processing portion of the Appendix). After combining the datasets, we examined stem/fact pairs in the dataset to ensure that we were feeding in “correct” factual associations to our tested models.This is work we felt important to carry out since, after all, the source datasets were programmatically scraped off of Wikidata. We randomly sampled a 25 rows subset of each stem/fact pair from each relation in the 33,870 row unfiltered dataset. We found that the stem/fact pairs linked, for example, via “citizen of”’ (relation P27) and “sister city”’ (relation P190) presented consistent inaccuracies when we performed manual verifications. We comprehensively removed these stem/fact pairs by filtering out dataset rows that are linked via that ‘relation’ (one of Wikidata’s metadata fields). **We can certainly include examples of stem/fact pairs we removed in the paper to illustrate why we believe their inclusion would detract from our benchmark’s viability and relevance.** This information, along with the specific number of rows removed that we already report in the Appendix, should make it eminently clear to the reader the exact process that we followed.
>
> **Lastly, thank you for pointing out the quick change we can make to large numbers (adding commas) to make them more legible. We will add this to our paper.**
>
> We hope that this rebuttal clarified some of the strengths of our project and demonstrated our intention to mend any potential shortcomings therein.

---

### Meta-Review · Area_Chair_yNw2 · 2023-09-19

**Recommendation:** 5

**Metareview:**

The reviewers agreed that this was an interesting study and the focus on non-English settings will have an impact on many people in the field working with non-English data. Even in a short paper, the authors make several important contributions, including a new multilingual dataset and evaluation of several major LLMs over the data with some interesting implications about the use of models in multilingual settings. There were no addressable reasons to reject pointed out other than lack of depth of some discussions; I do not see any major issues with the soundness of the paper that were pointed out. Several of the reviewers also found the paper to be an exciting direction. There were a few minor presentation and formatting suggestions for the authors to consider, as well.

---

### Decision · Program_Chairs · 2023-10-07

**Decision:**

Accept-Main

**Comment:**

The reviewers agreed that this was an interesting study and the focus on non-English settings will have an impact on many people in the field working with non-English data. Even in a short paper, the authors make several important contributions, including a new multilingual dataset and evaluation of several major LLMs over the data with some interesting implications about the use of models in multilingual settings. There were no addressable reasons to reject pointed out other than lack of depth of some discussions; I do not see any major issues with the soundness of the paper that were pointed out. Several of the reviewers also found the paper to be an exciting direction. There were a few minor presentation and formatting suggestions for the authors to consider, as well.